# THE BLACK–WHITE-BOX OPTIMIZATION NETWORK

## ABSTRACT

We introduce a *Black–White-Box Optimization Network* and its first instance, *Tensor-Train Creator (TTC)*, which couples Ising-style solves, a factorization-machine surrogate, and tensor-train (PROTES) search. Typed couplings, lattice realignment, and warm starts cut oracle calls and time-to-target. On black-box benchmarks and Max-Cut, TTC attains better values under the same evaluation budgets.

## 1 INTRODUCTION

Optimization "in the wild" mixes discrete choices, hidden constraints, simulator noise, and only partially known structure. *White-box* methods exploit available structure and (possibly implicit) gradients but struggle when objectives are non-differentiable or partly unknown; *black-box* methods are flexible but often sample-hungry. We propose the **Black–White-Box Optimization Network**, a modular framework that *interleaves* white-box solvers and black-box optimizers so that each fills the other's weak points via explicit, typed couplings.

We consider problems of the form

$$\min_{x \in \mathcal{X}} \ f(x) \quad \text{s.t.} \quad h(x) = 0, \ g(x) \leq 0, \tag{1}$$

where $x$ may be hybrid (e.g., $x = (b, \theta)$ with $b \in \{0,1\}^n$, $\theta \in \mathbb{R}^m$), parts of $h, g$ can be implicit or simulator-defined, and oracle evaluations of $f$ are expensive and noisy.

**A first instantiation:** Our first Black–White-Box Optimization Network architecture—**Tensor-Train Creator (TTC)**—composes three building blocks: (i) an **Ising-machine** path (e.g., quantum/coherent annealers) as a white-box route for structured QUBO-like subproblems Yamamoto et al. (2017); dwa (2020); Osaba & Miranda-Rodríguez (2024); (ii) **Higher-Order Factorization Machines (HOFM)** to expose tractable low-rank polynomial structure Blondel et al. (2016); and (iii) a **Tensor-Train (TT)** based black-box search layer such as **PROTES** Batsheva et al. (2023); Oseledets (2011). Concretely, the white path solves binary subproblems

$$\min_{b \in \{0,1\}^n} \ b^\top Q \, b + c^\top b, \tag{2}$$

yielding *white-box seeds* that bias the TT sampler. HOFM provides a compact surrogate

$$\hat{f}(x) = w_0 + \sum_i w_i x_i + \sum_{k=2}^{K} \sum_{i_1 < \cdots < i_k} \left\langle v_{i_1}^{(k)}, \ldots, v_{i_k}^{(k)} \right\rangle \prod_{j=1}^{k} x_{i_j}, \tag{3}$$

highlighting interactions amenable to Ising-structured solves and guiding variable grouping for the search layer.

**What changes in practice.** Under expensive evaluations, Black–White-Box Optimization Network/TTC aims to reduce overall complexity: fewer costly oracle calls and lower time-to-target than (i) pure black-box search, (ii) pure white-box solves, or (iii) a single grey-box surrogate. We also outline how to estimate potential quantum advantage at the *systems* level—i.e., when Ising-based seeding materially lowers the number of expensive evaluations on gradient-free hard problems dwa (2020); Osaba & Miranda-Rodríguez (2024).

**Contributions.** (1) We formalize the **Black–White-Box Optimization Network** as a typed graph of solver nodes (black/white) with explicit couplings; (2) we present Tensor Train Creator (TTC) model, the first Black–White-Box Optimization Network architecture combining Ising-machine white solves, HOFM structure, and TT/PROTES black-box search; (3) we catalogue *architecture options* (alternative seeds, priors, and coupling operators) without changing the core graph; and (4) we provide evidence that *overall complexity decreases*, with ablations isolating the value of seeding and white-path updates.

**Relation to prior work.** Two-module pipelines such as **FMQA** (Factorization Machines + Quantum Annealing) motivate this design, but TTC generalizes the idea to a *network* with explicit, iterative couplings and distributional control over exploration–exploitation, accommodating hybrid variables, hidden constraints, and simulator noise Kitai et al. (2020); Endo (2025).

## 2   RELATED WORK

Bayesian Optimization of Function Networks (BOFN) models intermediate nodes and selects evaluations that exploit network structure Astudillo & Frazier (2021); Buathong et al. (2024). Our *Black–White-Box Optimization Network* retains this *networked* perspective while mixing *heterogeneous solver types* (white ↔ black), supporting typed couplings, and permitting recursion. In parallel, differentiable optimization layers such as OptNet enable implicit/argmin differentiation through solvers, which allows gradient-based "white-path" updates even when the outer objective is gradient-free Amos & Kolter (2017).

A complementary line of work orchestrates quantum and classical components: Ising machines and hybrid approaches—including Coherent Ising Machines and D-Wave's hybrid solvers—tackle combinatorial problems via cross-paradigm coordination Yamamoto et al. (2017); dwa (2020); Osaba & Miranda-Rodríguez (2024); we formalize such coordination as a *typed solver network* with explicit module interfaces. Recent two-module pipelines (e.g., FMQA) couple a factorization-machine surrogate with quantum annealing in a tight learn–propose loop Kitai et al. (2020); Endo (2025), while PROTES performs black-box optimization via tensor-train (TT) sampling to explore massive discrete spaces efficiently Batsheva et al. (2023); Oseledets (2011).

Our *TTC* framework generalizes these ideas from two-node loops to *multi-node* typed networks with principled options for seeding and mixing, and it enables white-path updates wherever nodes are differentiable.

## 3   THE TENSOR-TRAIN CREATOR (TTC) OPTIMIZER

We consider discrete, gradient-free black-box optimization

$$x^\star \in \arg\min_x f(x), \qquad x = [n_1, \ldots, n_d], \quad n_i \in \{1, 2, \ldots, N_i\}. \tag{4}$$

Before introducing TTC, we analyze *PROTES*, the TT-based component our method builds upon, clarifying both its *black-box* usage (treating $f$ as an oracle) and its *white-box* leverage (injecting known structure directly into the TT sampler).

### 3.1   A TT-PARAMETERIZED SAMPLING DISTRIBUTION

PROTES (will be discussed in the next section) maintains a nonnegative *probability tensor* $P$ (possibly unnormalized) whose entries define a sampling distribution over multi-indices. The tensor $P$ is stored in the *tensor-train* (TT) format Oseledets (2011):

$$P[n_1, \ldots, n_d] = \sum_{r_1=1}^{R_1} \cdots \sum_{r_{d-1}=1}^{R_{d-1}} \mathcal{G}_1[1, n_1, r_1] \, \mathcal{G}_2[r_1, n_2, r_2] \cdots \mathcal{G}_d[r_{d-1}, n_d, 1], \tag{5}$$

with TT-cores $G_i \in \mathbb{R}^{R_{i-1} \times N_i \times R_i}$ and $R_0 = R_d = 1$. The first and last tensor cores are matrices (second order tensors) while the rest of cores are third-order tensors as shown in Figure 1. For uniform rank $R$, the parameter count is $O(d\,N\,R^2)$, where $N$ is a typical mode size.

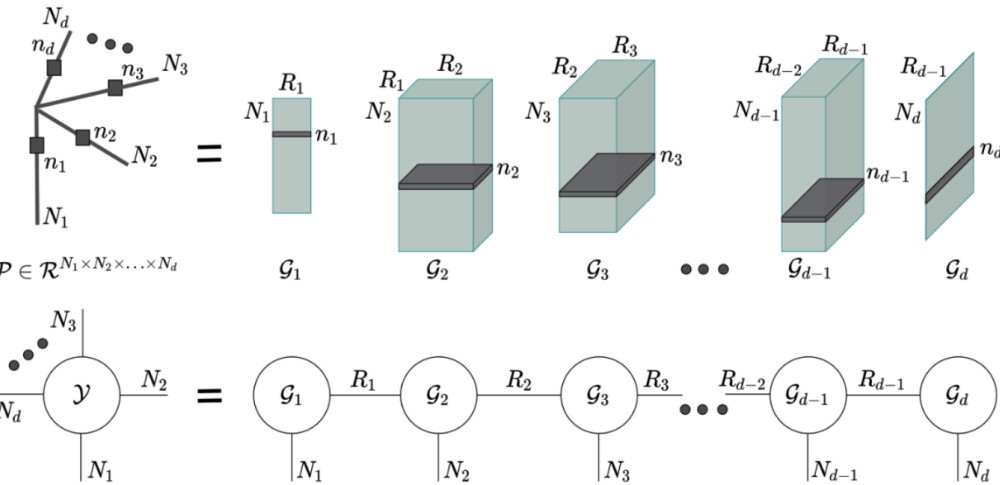

Figure 1: How the TT format works. Top: compute one entry $x[n_1, \ldots, n_d]$. Bottom: the same content as a tensor network.

In the TT format, computing $\log P[x]$ for a given multi-index $x$ costs $O(dR^2)$ (sequential $R \times R$ matrix transfers). Additionally, one can sample from the *unnormalized* distribution $P$ using a sequential-conditional TT sampler with time complexity

$$O\big(K\, d\, \big((N+R)R + \alpha(N)\big)\big) \tag{6}$$

where $\alpha(N)$ is the cost of drawing from a categorical distribution with $N$ outcomes. Both complexity results are established in Sec. 3 of the original work.

Figure 2 (Adapted from Batsheva et al. (2023), p. 3) sketches the full optimization pipeline.

## 3.2 PROTES: MODEL, ROLES, COMPLEXITY, AND BOTTLENECKS

The probability tensor $P$ serves as a probability model over the search space, where each entry corresponds to the likelihood of sampling a particular combination of discrete variable values. The idea of compactly representing a multivariable probability distribution in the TT format was first proposed in Dolgov et al. (2020). The PROTES algorithm uses this compact representation and iteratively refines $P$ by increasing probabilities for regions containing good solutions while decreasing them for poor regions, effectively concentrating sampling mass near optima. Crucially, $P$ is stored in the tensor-train (TT) format, which provides efficient compression for high-dimensional spaces while enabling efficient sampling and updates through sequential core operations. This TT-parameterized distribution allows PROTES to navigate complex discrete optimization landscapes without requiring gradient information from the black-box objective function.

At a high level, PROTES learns a TT-structured sampler $P$ that concentrates mass near minimizers. One iteration proceeds as follows:

   (i) Sample $K$ candidates from $P$;

  (ii) Evaluate $f$ at these candidates;

 (iii) Keep the top-$k$ indices by objective value;

 (iv) Update $P$ by maximizing their log-likelihood with $k_{\mathrm{gd}}$ Adam steps using automatic differentiation of $\log P[\cdot]$ through the TT cores.

Formally, with elite index set $S = \{s_1, s_2, \ldots, s_k\}$ and candidates $\{x_j\}_{j=1}^{K}$,

$$\mathcal{L}(P; \{x_{s_i}\}) = \sum_{i=1}^{k} \log P[x_{s_i}], \tag{7}$$

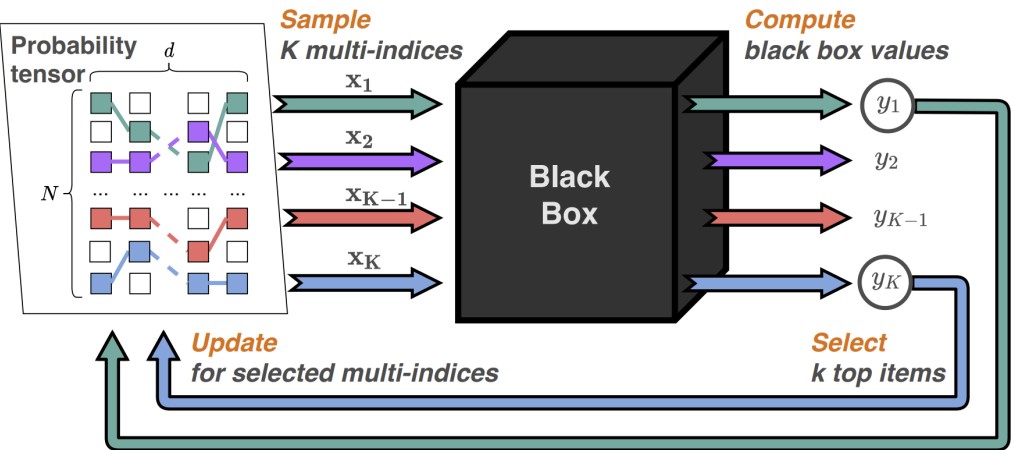

Figure 2: Conceptual outline of the PROTES optimization approach - Adapted from Batsheva et al. (2023), p. 3

which is maximized with respect to the TT parameters for $k_{\text{gd}}$ steps per iteration. Although PROTES is gradient-free with respect to $f$, its update can be connected to the REINFORCE identity by applying a monotone transformation (e.g., Fermi–Dirac) to $f$. In the low-temperature limit, only elite samples contribute, yielding equation 7. Empirically, using a single hyperparameter setting ($K = 100$, $k = 10$, $k_{\text{gd}} = 1$, $\lambda = 0.05$, $R = 5$) across 20 diverse benchmarks leads to the best value on 19 out of 20 problems.

IS PROTES BLACK-BOX OR WHITE-BOX?

PROTES is *black-box with respect to the evaluator $f$*, because it never computes $\nabla f$ or differentiates through $f$; it only uses the scalar values $f(x)$ to select an elite set $S$.

At the same time, each iteration solves a *white-box, differentiable internal subproblem* over the TT parameters $\theta$:

$$\mathcal{L}(\theta) = \sum_{x \in S} \log P_\theta(x), \qquad \nabla_\theta \mathcal{L}(\theta) = \sum_{x \in S} \nabla_\theta \log P_\theta(x),$$

where $P_\theta$ is represented in Tensor-Train (TT) form and gradients are obtained by automatic differentiation through the TT cores. Thus, PROTES performs *gradient-based optimization*, but *only* on its *own model $P_\theta$* (white-box), not on $f$ (which remains black-box).

Table 1: PROTES roles with respect to the *true* objective $f$ and the *internal* sampler $P_\theta$.

| Aspect | Role | What it means in PROTES |
|---|---|---|
| Evaluator $f$ | Black-box | No $\nabla f$, no backprop through $f$; $f(x)$ is used only to rank $K$ candidates and pick the elite set $S$. |
| Sampler update ($P_\theta$) | White-box | Optimize $\theta$ by maximizing $\sum_{x \in S} \log P_\theta(x)$ with gradients $\nabla_\theta \log P_\theta(x)$ through TT cores (AD). |
| Known structure | White-box (optional) | Constraints/priors can be encoded directly into $P_\theta$ (e.g., TT indicators, structured cores), reducing wasted queries. |

**Internal complexity and wall time.** Let $M$ be the number of evaluations of $f$ (budget). Since each iteration evaluates $K$ samples, the number of iterations is $M/K$.

**Theorem 1** (PROTES internal time)**.** *Excluding the cost of evaluating $f$, one PROTES run performs*

$$T_{\text{int}} = O\Big(M\, d\big((N+R)R + \alpha(N)\big) + M\, d\, \tfrac{k}{K}\, k_{\text{gd}}\, R^2\Big), \tag{8}$$

*where $\alpha(N)$ is the cost of sampling a categorical with $N$ outcomes.*

*Proof.* Sequential-conditional TT sampling costs $O(K\,d((N+R)R + \alpha(N)))$ per iteration; over $M/K$ iterations this yields the first term. Computing $\log P_\theta(x)$ and its gradient is $O(dR^2)$ per elite; over $k_{\mathrm{gd}}$ steps and $k$ elites for $M/K$ iterations gives the second term. See App. A for details. $\qquad\square$

Total wall time decomposes as

$$T_{\mathrm{total}} = M\,T_f + T_{\mathrm{int}}, \qquad (9)$$

with $T_f$ the average per-call latency of $f$. Two practical bottlenecks follow: (i) if $f$ is expensive, $M\,T_f$ dominates; (ii) the outer loop is *sequential* across iterations because the update at $t+1$ depends on elites from $t$ (even though the $K$ evaluations within each iteration can be run in parallel).

## 4 WHY DENSE QUBO IS HARD AND HUBO CAN BE EASIER

Empirically, reports by Batsheva et al. (2023) and follow-up experiments (e.g., Salloum 2025; Salloum et al. 2025) observe degradation on dense QUBO as dimension grows, while converting QUBO→HUBO can improve outcomes at fixed dimension. A common misinterpretation is that "TT is better on higher-order tensors." In contrast, our analysis shows that the decisive factor is *lattice alignment*: TT/MPS complexity is governed by the size of graph/hypergraph separators along the 1D ordering, not by the algebraic order of terms.

**Theorem 2** (Lattice alignment lower bound). *Let $\mu(x) \propto e^{-\beta E(x)}$ be the Gibbs distribution of energy $E$ on a graph/hypergraph $G$ over variables $(n_1, \ldots, n_d)$, and let $\pi$ be a 1D ordering. For every $i \in \{1, \ldots, d-1\}$, the TT bond dimension $R_i$ required to represent $\mu$ exactly along $\pi$ satisfies*

$$R_i = \mathrm{rank}\big(\mathrm{Unfold}_i(\mu)\big) \geq 2^{\Omega(\,\mathrm{tw}_\pi(G,i)\,)},$$

*where $\mathrm{tw}_\pi(G,i)$ is the size of the minimum separator induced by the cut $(\{\pi(1), \ldots, \pi(i)\}, \{\pi(i+1), \ldots, \pi(d)\})$. Moreover, the same separator controls an upper bound: $R_i \leq 2^{\mathrm{tw}_\pi(G,i)}$. Consequently, if $G$ has large separators for all 1D orderings (e.g., fully connected or 2D grids), the minimal ranks grow super-polynomially with $d$; if $G$ is pruned to a near-chain topology, ranks remain small.*

*Proof.* In TT/MPS, the bond dimension $R_i$ equals the Schmidt rank across the bipartition at position $i$. The Schmidt rank lower-bounds exponentially in the number of independent constraints crossing the cut. For graphical models, the number of such independent constraints scales with the size of a minimum vertex separator across the cut, i.e., the (pathwise) treewidth along $\pi$. This yields the exponential lower bound; see App. B and standard TT/MPS expressivity results Oseledets (2011). The upper bound follows by constructing a junction-tree-like contraction across a separator of size $\mathrm{tw}_\pi(G,i)$, implying $R_i \leq 2^{\mathrm{tw}_\pi(G,i)}$. $\qquad\square$

**Corollary (Dense QUBO is hard).** Let $G$ be the interaction graph of a QUBO on $d$ variables. If $G$ is dense (e.g., $G = K_d$), then for any ordering $\pi$ there exists an $i$ with $\mathrm{tw}_\pi(G,i) = \Omega(d)$, hence by Theorem 2

$$R_i \geq 2^{\Omega(d)}.$$

Therefore, exact TT representations of $\mu$ require exponentially large bond dimensions across at least one cut for every $\pi$, making dense QUBO intractable for TT-based methods.

*Proof sketch.* For $G = K_d$, every cut after position $i$ has all $i(d-i)$ cross-edges. Any separator must involve at least $\min\{i, d-i\}$ vertices, so $\mathrm{tw}_\pi(K_d, i) = \Omega(\min\{i, d-i\})$. Taking $i = \lfloor d/2 \rfloor$ yields $\Omega(d)$, and Theorem 2 completes the claim. $\qquad\square$

**Proposition (HUBO can be easier via lattice alignment).** There exist QUBO→HUBO transformations that preserve the minimizers (and approximate the Gibbs measure on the feasible manifold) while producing a hypergraph $H$ that admits a near-chain ordering $\pi$ with $\mathrm{tw}_\pi(H, i) = O(w)$ for some small window $w$. Consequently, by Theorem 2 all TT ranks satisfy $R_i \leq 2^{O(w)}$, i.e., are polynomially bounded when $w$ is constant or slowly growing.

*Construction proof sketch.* Fix a target ordering $\pi$ and partition variables into contiguous blocks $B_1, \ldots, B_m$ of size at most $w$. Introduce auxiliary summary bits $s_b$ that encode prescribed local statistics of $x_{B_b}$, and replace dense cross-block pairwise terms by higher-order *local* HUBO terms within blocks plus (optionally) interactions between adjacent summaries $(s_b, s_{b+1})$. Enforce $s_b$-$x_{B_b}$ consistency with HUBO penalties confined to $B_b$. The resulting hypergraph $H$ has hyperedges only within blocks or between adjacent blocks, so every cut intersects $O(w)$ variables plus $O(1)$ summaries. Hence $\mathrm{tw}_\pi(H, i) = O(w)$ and $R_i \leq 2^{O(w)}$ by Theorem 2. □

**Examples and sanity checks.** (i) *All-to-all but low intrinsic rank.* Energies depending only on $\sum_i x_i$ (Curie–Weiss-type) are pairwise dense but factor through a global summary. Encoding that summary as a single higher-order term aligned with $\pi$ yields $\mathrm{tw}_\pi = O(1)$, so HUBO has small TT ranks. (ii) *Local $k$-HUBO chains.* If each term touches variables within a sliding window of width $w$ along $\pi$, then $\mathrm{tw}_\pi = O(w)$ and $R_i \leq 2^{O(w)}$.

**Conclusion (why dense QUBO is hard and HUBO can be easier).** TT complexity is dictated by separator sizes of the *interaction lattice* along the chosen ordering, not by whether the energy is quadratic or higher-order. Dense QUBO induces large separators for all orderings, forcing $R_i \geq 2^{\Omega(d)}$ and yielding super-polynomial scaling. Converting QUBO→HUBO can realign interactions into a near-chain hypergraph with small separators, giving $R_i \leq 2^{O(w)}$ and tractable TT compression. This explains the empirical pattern reported in Batsheva et al. (2023) and follow-up work.

**Path dependence.** Initialization matters because the elite-likelihood ascent increases a variational lower bound.

**Theorem 3** (Initialization controls sample complexity). *Fix an ordering $\pi$ and rank profile $(R_1, \ldots, R_{d-1})$, and let $\widetilde{P}$ be the best TT approximation to $\mu(x) \propto e^{-\beta f(x)}$ under $(\pi, R)$. If the initial sampler $P_{\theta_0}$ satisfies $D_{\mathrm{KL}}(\widetilde{P} \| P_{\theta_0}) = \Delta_0$, then elite-likelihood ascent with step size $\eta$ and elite ratio $k/K$ reduces the gap as*

$$\mathbb{E}\big[\Delta_{t+1} \,\big|\, \Delta_t\big] \;\leq\; \Delta_t\Big(1 - c\,\eta\,\tfrac{k}{K}\Big),$$

*for a constant $c > 0$ depending on local smoothness and mixing, so the expected iteration count to reach $P_\theta(x^\star) \geq 1 - \varepsilon$ is $O(\Delta_0/(\eta\, k/K))$.*

*Proof.* Write $L(\theta) = \sum_{x \in S} \log P_\theta(x)$ and note that $D_{\mathrm{KL}}(\widetilde{P} \| P_\theta) = \mathrm{const} - \mathbb{E}_{\widetilde{P}}[\log P_\theta(x)]$ up to approximation error. A standard smoothness argument for stochastic ascent on $L$ with elite sampling yields linear contraction in expectation; see App. C. □

Theorems 2–3 explain the QUBO/HUBO observations: converting some dense QUBO instances to HUBO can *reduce* separators (hence ranks) by inducing a near-chain hypergraph; the higher algebraic order does not harm TT if interactions remain *local* along the chain. Conversely, simply increasing ranks cannot overcome dense long-range couplings without reshaping the lattice.

### 4.1 THE TTC ARCHITECTURE

The *Tensor-Train Creator* (TTC) addresses two fundamental bottlenecks identified in 2–3: (i) TT ranks blow up on non–chain-like lattices, and (ii) elite-likelihood ascent is path dependent and sensitive to initialization. TTC remedies these issues by combining a higher-order factorization machine (HOFM) surrogate to expose structure, an *Ising/annealing* stage to warm-start and *shape* the surrogate lattice, and an *annealed* PROTES loop that updates the TT sampler on the true objective. The

---

**Algorithm 1** TTC: Tensor-Train Creator

---

**Require:** budget $M$; initial batch $B_0$; samples/iter $K$; elites $k$; initial order $m \leftarrow 2$; annealing config (temps, reads), cooling schedule $\tau \downarrow 0$
1: **Seed:** draw $B_0$ points, evaluate $f$ in parallel; $\mathcal{D} \leftarrow \{(x, f(x))\}_1^{B_0}$
2: **while** $(|\mathcal{D}| < M)$ **and** (not converged) **do**
3:     (**Surrogate**) train HOFM $\hat{f}_m$ on $\mathcal{D}$ (few epochs)
4:     (**White-box**) build QUBO $E_{\text{sur}}$ from $\hat{f}_m$ (always we take the quadratic part of HOFM - $m$=2)

5:     (**Anneal**) solve $E_{\text{sur}}$; obtain best $\hat{z}$ and edge stabilities E
6:     (**Shape**) prune weak couplings using E; contract consistent chains; get $G'$
7:     (**TT-Create**) seriate $G'$ to get ordering $\pi$; estimate ranks $(R_i)$; init $P_\theta$ by moment matching
8:     **for** $\mathtt{t} = \mathtt{1,2,...}$ **while** budget remains and validation improves **do**
9:        sample $K$ points $X_t \sim P_\theta^{(\tau_t)}$; evaluate $Y_t = \{f(x) : x \in X_t\}$ in parallel
10:       select elites $S_t = \arg\min_{x \in X_t}$ (top-$k$ by $f$); update $\theta \leftarrow \theta + \eta \nabla_\theta \sum_{x \in S_t} \log P_\theta(x)$
11:       $\mathcal{D} \leftarrow \mathcal{D} \cup \{(x, f(x)) : x \in X_t\}$; $\tau_{t+1} \leftarrow \gamma_\tau \tau_t$
12:       **if** (retrain period) **then** update $\hat{f}_m$ on current $\mathcal{D}$ **end if**
13:     **end for**
14:     **if** (validation plateau) **then** $m \leftarrow m + 1$ **end if**
15: **end while**
16: **return** $\arg\min_{(x,y) \in \mathcal{D}} y$

---

lattice is reshaped by restricting the HOFM surrogate to the relevant conditions, and the second issue is mitigated via annealing-based pruning. The ways of pruning via annealing-based solver (Ising machine) is shown in appendix E.

**Pipeline.** Given a budget of $M$ black-box calls, TTC maintains a dataset $\mathcal{D}$, a restricted HOFM surrogate $\hat{f}_m$ of order $m$, a shaped interaction graph $G'$, and a TT sampler $P_\theta$ over $\mathcal{X} = \prod_i [N_i]$. A single macro-cycle consists of:

1. **Parallel seeding**: evaluate an initial batch $B_0$ of points to form $\mathcal{D}_0 = \{(x, f(x))\}$.

2. **HOFM (order $m$)**: fit $\hat{f}_m$ with ANOVA-style factorization and chain-aware penalties; complexity per epoch is $O(m\, d\, k_{\text{H}}\, |\mathcal{D}|)$, where $k_{\text{H}}$ is the HOFM rank parameter Blondel et al. (2016).

3. **Surrogate → QUBO (white-box)**: take the $m = 2$ part from HOFM i.e. FM.

4. **Annealing (warm-start & shaping)**: solve the QUBO via an Ising backend (quantum, quantum-inspired, or GPU) to obtain solutions $\hat{z}$ and edge stabilities. Use these to (a) *warm-start* the TT by moment matching of low-order marginals (reduces the initial KL gap $\Delta_0 \mapsto \Delta_1$; 3), and (b) *prune/contract* to produce a near-chain $G'$ (shrinks effective ranks; 2).

5. **TT creation & annealed PROTES**: compute an ordering $\pi$ by seriation on $G'$, estimate a rank profile $(R_i)$ from surrogate matricizations (App. D), initialize $P_\theta$, and run an *annealed* PROTES loop on the true $f$: sample $K$ points from $P_\theta^{(\tau)} \propto P_\theta^{1/\tau}$, evaluate $f$ in parallel, update $\theta$ with the elite objective, augment $\mathcal{D}$, and periodically retrain $\hat{f}_m$. Increase $m$ only when validation saturates (order curriculum).

### 4.1.1 SPEED-UPS AND WHEN THEY APPEAR

Let $T_{\text{TT}}(R)$ denote the internal TT cost per $f$-evaluation (cf. equation 8). If annealing reduces the initialization gap by $S_{\text{init}} := \Delta_0/\Delta_1$ and the effective ranks by $\gamma \in (0, 1)$ (so $R \mapsto \gamma R$), and if the annealer overhead per macro-cycle is $C_{\text{Ising}}$, then the internal speed-up (same black-box budget $M$) is

$$S_{\text{int}} \approx \frac{T_{\text{TT}}(R)}{(\Delta_1/\Delta_0)\, T_{\text{TT}}(\gamma R) + C_{\text{Ising}}/M} \approx \frac{1}{(\Delta_1/\Delta_0)\, \gamma^2 + C_{\text{Ising}}/(M\, T_{\text{TT}}(R))}, \quad (10)$$

since the dominant TT terms scale as $O(dR^2)$. This aligns with 3 –2.

## 4.2 COMPLEXITY ANALYSIS

We summarize the dominant work (excluding constant factors). Let $N = \max_i N_i$; $R$ be the nominal TT rank budget in PROTES; $\bar{R}$ be the TTC *effective* rank after shaping; $d$ be the dimension; $K$ the samples per iteration; $k$ the elite count; $k_{\mathrm{gd}}$ the ascent steps per iteration; $T = |\mathcal{D}|$ the dataset size at a given time; $m$ the HOFM order; $k_{\mathrm{H}}$ the HOFM rank; and $\alpha(N)$ the cost of sampling a categorical ($\alpha(N) \le N$ naively).

**PROTES (for comparison).** Internal time for a budget $M$ is

$$T_{\mathrm{int}}^{\mathrm{PROTES}} = O\Big( M\, d\big((N+R)R + \alpha(N)\big) + M\, d\, \tfrac{k}{K}\, k_{\mathrm{gd}}\, R^2 \Big), \tag{11}$$

and total wall-time $T_{\mathrm{tot}}^{\mathrm{PROTES}} = M\, T_f + T_{\mathrm{int}}^{\mathrm{PROTES}}$.

**FMQA (for comparison).** Per outer iteration: (i) fit FM/HOFM on $\mathcal{D}$ in $O(m\, d\, k_{\mathrm{H}}\, T)$ per epoch; (ii) solve the surrogate by annealing with overhead $C_{\mathrm{Ising}}$; (iii) evaluate a small batch on $f$. If $E_m$ epochs and $B_{\mathrm{fm}}$ new points per iteration, then over $M$ evaluations (about $M/B_{\mathrm{fm}}$ iterations):

$$T_{\mathrm{int}}^{\mathrm{FMQA}} = O\Big( \underbrace{\textstyle\sum_{\mathrm{iters}} E_m\, m\, d\, k_{\mathrm{H}}\, T}_{\text{surrogate training}} + \underbrace{(M/B_{\mathrm{fm}})\, C_{\mathrm{Ising}}}_{\text{annealing solves}} \Big), \quad T_{\mathrm{tot}}^{\mathrm{FMQA}} = M\, T_f + T_{\mathrm{int}}^{\mathrm{FMQA}}. \tag{12}$$

FMQA has no TT terms but relies more heavily on annealing and surrogate quality.

**TTC (this work).** TTC's internal cost over a budget $M$ decomposes into

$$\text{TT sampling}: \quad O\Big( M\, d\big((N+\bar{R})\bar{R} + \alpha(N)\big) \Big), \tag{13}$$

$$\text{TT updates}: \quad O\Big( M\, d\, \tfrac{k}{K}\, k_{\mathrm{gd}}\, \bar{R}^2 \Big), \tag{14}$$

$$\text{HOFM training}: \quad O\Big( \sum_{\text{macro-cycles}} E_m\, m\, d\, k_{\mathrm{H}}\, T \Big), \tag{15}$$

$$\text{Annealing \& shaping}: \quad O\Big( \sum_{\text{macro-cycles}} C_{\mathrm{Ising}} \Big), \tag{16}$$

with $\bar{R} = \gamma R$ and an *iteration* contraction factor of about $\Delta_1/\Delta_0$ from the warm-start (3). Aggregating:

$$T_{\mathrm{int}}^{\mathrm{TTC}} = O\Big( M\, d\big((N+\bar{R})\bar{R} + \alpha(N)\big) + M\, d\, \tfrac{k}{K}\, k_{\mathrm{gd}}\, \bar{R}^2 + \textstyle\sum_{\text{cycles}} E_m\, m\, d\, k_{\mathrm{H}}\, T + \sum_{\text{cycles}} C_{\mathrm{Ising}} \Big). \tag{17}$$

Total wall-time:

$$T_{\mathrm{tot}}^{\mathrm{TTC}} = M\, T_f + T_{\mathrm{int}}^{\mathrm{TTC}}. \tag{18}$$

**Interpretation.** Compared to equation 11, TTC replaces $R$ by $\bar{R} = \gamma R$ in all TT terms and benefits from faster concentration in practice (fewer effective PROTES iterations to a target quality by 3). The additional costs are the *surrogate* terms equation 15 and *annealing* terms equation 16. When (i) $M$ is moderate/large; (ii) shaping is effective ($\gamma \ll 1$); and (iii) $C_{\mathrm{Ising}}$ is amortized (batched reads), the reduction in the dominating $O(dR^2)$ TT terms outweighs surrogate/annealing overheads, yielding the internal speed-up in equation 10.

## 4.3 COMPLEXITY COMPARISON

Let $\Theta_{\mathrm{TT}}(R) := d\big((N+R)R + \alpha(N)\big) + d\, \tfrac{k}{K}\, k_{\mathrm{gd}}\, R^2$ denote the per-evaluation TT work. The principal internal and total costs are:

| Method | Internal time | Total time |
|---|---|---|
| PROTES | $M\, \Theta_{\mathrm{TT}}(R)$ | $M\, T_f + M\, \Theta_{\mathrm{TT}}(R)$ |
| FMQA | $\sum E_m\, m\, d\, k_{\mathrm{H}}\, T + (M/B_{\mathrm{fm}})\, C_{\mathrm{Ising}}$ | $M\, T_f + \sum E_m\, m\, d\, k_{\mathrm{H}}\, T + (M/B_{\mathrm{fm}})\, C_{\mathrm{Ising}}$ |
| TTC (ours) | $M\, \Theta_{\mathrm{TT}}(\bar{R}) + \sum E_m\, m\, d\, k_{\mathrm{H}}\, T + \sum C_{\mathrm{Ising}}$ | $M\, T_f + M\, \Theta_{\mathrm{TT}}(\bar{R}) + \sum E_m\, m\, d\, k_{\mathrm{H}}\, T + \sum C_{\mathrm{Ising}}$ |

Here $\bar{R} = \gamma R$ with $\gamma \in (0,1)$ determined by lattice shaping; the *iteration* count to reach a target is reduced proportional to $\Delta_0/\Delta_1$ by warm-starting (3). Thus TTC's internal advantage over PROTES scales like $\gamma^{-2}$ (rank shrink) multiplied by $S_{\mathrm{init}} = \Delta_0/\Delta_1$ (fewer effective TT iterations), up to the additive surrogate/annealing terms.

**When TTC wins.** If $M$ is dominated by TT work (cheap $f$) and $\gamma \ll 1$, the $\bar{R}^2$ term reduction drives clear wins. If $f$ is expensive, all methods share the same $M\,T_f$, so the method that reduces *required $M$ to reach a target* wins; TTC does so via better initialization and lattice alignment.

ESTIMATING SYSTEMS-LEVEL QUANTUM ADVANTAGE FROM ISING-BASED SEEDING

Consider a baseline run requiring $M_0$ oracle evaluations with per-call latency $T_f$. Let the TT internal work per evaluation be

$$\Theta_{\mathrm{TT}}(R) := d\big((N+R)R + \alpha(N)\big) + d\,\frac{k}{K}\,k_{gd}\,R^2.$$

With Ising-based seeding, suppose the evaluation count shrinks by a factor $\rho \in (0,1)$ and effective ranks contract as $R \mapsto \bar{R} = \gamma R$ with $\gamma \in (0,1)$. Let $\overline{C}_{\mathrm{HOFM+Ising}}$ be the total overhead of surrogate training and seeding (including the annealer time-to-solution).

**Systems-level win condition.** TTC with Ising seeding improves total wall time over baseline whenever

$$M_0\Big[(1-\rho)\,T_f + \Theta_{\mathrm{TT}}(R) - \rho\,\Theta_{\mathrm{TT}}(\bar{R})\Big] \; > \; \overline{C}_{\mathrm{HOFM+Ising}} \tag{$\dagger$}$$

**Useful thresholds.** Define $\bar{c} := \overline{C}_{\mathrm{HOFM+Ising}}/M_0$.

$$T_f \; > \; T_f^{\star} := \frac{\bar{c} - \big(\Theta_{\mathrm{TT}}(R) - \rho\,\Theta_{\mathrm{TT}}(\bar{R})\big)}{1-\rho}, \qquad M_0 \; > \; M_{\min} := \frac{\overline{C}_{\mathrm{HOFM+Ising}}}{(1-\rho)\,T_f + \Theta_{\mathrm{TT}}(R) - \rho\,\Theta_{\mathrm{TT}}(\bar{R})}.$$

**Interpreting $\rho$ and $\gamma$.** Warm-start quality controls $\rho$ (fewer expensive evaluations); rank shaping controls $\gamma$ (cheaper internal work). A crude rule of thumb is $\Theta_{\mathrm{TT}}(\bar{R}) \approx \gamma^2\,\Theta_{\mathrm{TT}}(R)$, yielding extra headroom $(1-\rho\gamma^2)\Theta_{\mathrm{TT}}(R)$ to pay for seeding overheads.

**Quantum advantage.** If quantum seeding has lower total overhead than classical seeding, i.e. $\overline{C}^{(Q)}_{\mathrm{HOFM+Ising}} < \overline{C}^{(C)}_{\mathrm{HOFM+Ising}}$, and also satisfies ($\dagger$) with the same $\rho, \gamma$, then a *systems*-level quantum advantage is achieved by materially reducing the number of expensive oracle calls.

## 5 NUMERICAL RESULTS

Table 2: Results by problem and algorithm. Row minima (in TTC, except Schwefel) are in **bold**. Values use two decimals for $|x| \geq 10^{-2}$, and scientific notation with two significant digits for $|x| < 10^{-2}$.

| Problem / Algorithm | TTC | Protes | BS1 (DE) | BS2 (PSO) |
|---|---|---|---|---|
| ackley | **$4.44 \times 10^{-16}$** | 0.78 | $4.30 \times 10^{-3}$ | $9.10 \times 10^{-8}$ |
| rastrigin | **2.00** | 12.00 | 31.18 | 5.99 |
| griewank | **0.04** | 0.11 | 0.43 | 0.05 |
| rosenbrock | **1800.00** | 5942.00 | 2578.00 | 1897.00 |
| schwefel | **-402.00** | -401.20 | -398.00 | -399.50 |
| michalewicz | **-45.50** | -32.11 | -45.10 | -41.23 |
| levy | **5.90** | 10.39 | 7.84 | 5.98 |
| max-cut | **-12686.30** | -10367.80 | -11123.40 | -10012.90 |

## 6 CONCLUSION

TTC formalizes typed couplings and fixes TT bottlenecks, translating into faster convergence and fewer evaluations than PROTES and classical baselines. Limits include surrogate quality and annealer overhead; future work adds richer node types and sharper rank/ordering estimates.

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

## A PROTES: SAMPLING/UPDATE COSTS AND PROOFS

### A.1 SEQUENTIAL-CONDITIONAL TT SAMPLING

The sampler generates $x_1 \sim P(n_1)$, then $x_2 \sim P(n_2 \mid n_1)$, etc., using left/right *messages* computed by contracting TT cores. For each step, evaluating conditionals costs $O((N + R)R)$; sampling a categorical costs $\alpha(N)$. Over $K$ draws and $d$ dimensions, the sampling cost per iteration is $O(K\,d((N + R)R + \alpha(N)))$.

### A.2 GRADIENT OF THE ELITE LOG-LIKELIHOOD

For any index $x$, evaluating $\log P_\theta(x)$ is $O(dR^2)$ via forward contractions; reverse-mode backprop through cores is the same order. Over $k$ elites and $k_{\text{gd}}$ steps per iteration, the update cost is $O(k_{\text{gd}}\,k\,dR^2)$. Summing across $M/K$ iterations proves Theorem 1.

## B LATTICE ALIGNMENT: PROOFS OF THEOREMS 2

For an ordering $\pi$, the TT bond dimension $R_i$ equals the matrix rank of the unfolding across the cut $(\{1{:}i\}, \{i{+}1{:}d\})$. Interpreting $P_\theta$ as an MPS and $\mu$ as a target state, $R_i$ equals the Schmidt rank across that bipartition. In a graphical model on $G$, the conditional dependence across the cut is controlled by the size of a minimal separator $S_i$. If $|S_i|$ grows with $d$, then the number of independent constraints across the cut grows, implying $R_i \geq 2^{\Omega(|S_i|)}$ for exact representation. This gives Theorem 2.

## C INITIALIZATION AND KL CONTRACTION: PROOF OF THEOREM 3

Let $\widetilde{P}$ be the best TT approximation under $(\pi, R)$ and define $\Phi(\theta) = \mathbb{E}_{x \sim \widetilde{P}}[\log P_\theta(x)]$. Then $D_{\text{KL}}(\widetilde{P}\|P_\theta) = \text{const} - \Phi(\theta)$. Elite-likelihood ascent implements a noisy ascent on $\Phi$ (up to a temperature scaling) with step size $\eta$ and sampling ratio $k/K$. Assuming local smoothness and a Polyak-Łojasiewicz condition in a neighborhood of $\theta^\star = \arg\max \Phi$, the standard SGD recursion gives $\mathbb{E}[\Phi(\theta_{t+1}) - \Phi(\theta_t)] \geq c\,\eta\,\frac{k}{K}\left(\Phi^\star - \Phi(\theta_t)\right)$, for a constant $c > 0$, hence geometric contraction of the gap and the stated iterations bound.

## D RESTRICTED HOFM, QUBO EXTRACTION, AND TT CREATION

### D.1 HOFM TRAINING AND COMPLEXITY

The ANOVA trick Blondel et al. (2016) computes all orders $2{:}m$ interactions in $O(m\,d\,k)$ passes per sample using DP recurrences, yielding $O(m\,d\,k\,T)$ per epoch for dataset size $T$. Chain-aware group sparsity penalizes interactions far in the current ordering.

### D.2 SURROGATE → QUBO (WHITE-BOX)

For $m{=}2$, one-hot encodings yield a QUBO directly. For $m{>}2$, quadratize by introducing auxiliary variables $u$ and penalties $\lambda$ such that monomials $\prod_{i \in S} z_i$ are represented by $u_S$ with constraints $u_S = z_{i_1} \wedge \cdots \wedge z_{i_{|S|}}$, producing $E_{\text{sur}}(z, u) = z^\top Q z + \lambda\,\text{penalty}(z, u)$.

### D.3 ORDERING AND RANK ESTIMATION

Given a pruned graph $G'$, obtain an ordering $\pi$ via seriation (e.g., spectral ordering). Estimate a rank profile $(R_i)$ by inspecting singular spectra of surrogate matricizations across the cuts $i$ and selecting minimal $R_i$ meeting an energy threshold (Tikhonov-stabilized).

## D.4 MOMENT-MATCHING INITIALIZATION AND ANNEALED SAMPLING

From best surrogate solutions, build univariate and adjacent-pair marginals along $\pi$ and initialize TT cores to match them at moderate inverse temperature $\beta$. Use tempered TT sampling $P_\theta^{(\tau)} \propto P_\theta^{1/\tau}$ with $\tau \downarrow 0$ to balance exploration/exploitation.

# E ANNEALER-GUIDED CONVERSION OF A 2D QUBO TO A STRICT 1D NEAREST-NEIGHBOR QUBO

In this section, We present a practical pipeline that converts a quadratic unconstrained binary optimization (QUBO) defined on a 2D grid into a *strict 1D nearest-neighbor* (NN) QUBO suitable for line-structured Ising hardware. The method is *annealer-guided*: (i) obtain a reference optimum by annealing the original instance, (ii) *select* only the most valuable long-range couplers to embed exactly using equality-chain ancillas, under a user-specified ancilla budget, and (iii) prune or fold the remaining long-range interactions while preserving fidelity to the original optimum in both energy and configuration. We provide precise inputs/outputs, core identities, two selection strategies (greedy and one-shot knapsack), a penalty sweep to calibrate equality strength, and implementable pseudocode.

## E.1 PROBLEM STATEMENT, INPUTS, AND OUTPUTS

**Original QUBO.**
$$E(x) \;=\; x^t opQ\, x \;+\; q^t opx \;+\; c, \qquad x \in \{0, 1\}^n, \tag{19}$$
with $Q \in \mathbb{R}^{n \times n}$ (w.l.o.g. upper triangular, $Q_{ii} = 0$), $q \in \mathbb{R}^n$, and $c \in \mathbb{R}$, arranged on a 2D grid.

**Inputs.**

- QUBO coefficients $(Q, q, c)$ and variable→grid mapping.
- Access to an annealer ANNEAL$(Q, q, c; \cdot)$ returning best energy and sample.
- A line ordering $p : \{0, \ldots, n-1\} \to \{0, \ldots, n-1\}$ (e.g., snake/Hilbert).
- Ancilla budget $K$ (maximum total chain length allowed for embeddings).
- Equality-penalty seed $M_0$ and a small sweep set, e.g., $\{M_0, 2M_0, 4M_0\}$.
- Pruning policy for non-embedded long edges: `drop` or `fold`.

**Outputs.**

- A strict 1D NN QUBO $(Q', q', c')$ (including ancillas).
- Final annealed solution $(x_{\text{final}}, E_{\text{final}})$ on $(Q', q', c')$.
- Selected set $S \subseteq \mathbb{E}_{far}$ of long edges embedded exactly; total ancillas $\leq K$.

## E.2 PRELIMINARIES: LINE ORDER AND EDGE CLASSES

Let $p$ be the chosen permutation (grid→line) and define $pos(i) = p(i)$. Partition the quadratic terms:

$$= \{(i,j) : pos(i) - pos(j) = 1,\ i < j\}, \qquad \mathbb{E}_{far} = \{(i,j) : pos(i) - pos(j) > 1,\ i < j\}.$$

For a long-range ("far") edge $e = (i, j)$, the *chain length cost* (ancillas needed to route it along the line) is

$$L_e = pos(i) - pos(j) - 1. \tag{20}$$

## E.3 PRUNING POLICIES FOR LONG EDGES

For a far term $Q_{ij}x_i x_j$ we consider two simple policies:

- **drop:** remove the term.

- **fold** (energy-guided, using the reference optimum $x^*$):

$$Q_{ij} x_i x_j \approx Q_{ij}\Big(x_i\, x_j^* + x_j\, x_i^* - x_i^* x_j^*\Big). \tag{21}$$

This replacement adds linear terms in $x_i, x_j$ plus a constant, preserving strict 1D NN topology.

### E.4 EXACT 1D NN EMBEDDING VIA EQUALITY CHAINS

For a selected far edge $e = (i,j)$, let the line path be $i = v_0, v_1, \ldots, v_L = j$. Introduce ancillas $z_1, \ldots, z_{L-1}$ and enforce *equality* along adjacent nodes using the purely quadratic penalty:

$$(u,v) = M\,(u-v)^2 = M\,(u+v-2uv), \tag{22}$$

applied to links $(x_{v_0}, z_1), (z_1, z_2), \ldots, (z_{L-1}, x_{v_L})$. At the minimum (for $M$ large enough), $x_{v_0} = z_1 = \cdots = z_{L-1} = x_{v_L}$, so we can *replace* the far quadratic by a *local* linear term on any chain node:

$$Q_{ij}\, x_i x_j \longrightarrow Q_{ij}\, z_m, \qquad m \in \{1, \ldots, L-1\}. \tag{23}$$

All quadratics remain NN along the line; the model stays a pure QUBO.

### E.5 REFERENCE OPTIMUM BY ANNEALING

We first solve the original instance by annealing to obtain a target optimum:

$$x^* \in \underset{x \in \{0,1\}^n}{\arg\min} E(x), \qquad E^* = E(x^*). \tag{24}$$

This pair $(x^*, E^*)$ guides selection and (if used) folding equation 21.

### E.6 SELECTION STRATEGIES FOR LONG EDGES

We propose two annealer-guided strategies to pick a budgeted subset $S \subseteq \mathbb{E}_{far}$ to embed exactly.

#### E.6.1 GREEDY ANNEALER-GUIDED SELECTION (ITERATIVE)

Maintain a current embedded set $S$ and its best annealed energy $E_S$ for the strict 1D NN model that embeds $S$ (and prunes others by the chosen policy). For any candidate $e \in \mathbb{E}_{far} \setminus S$, let $E_{S \cup \{e\}}$ be the best annealed energy when $e$ is added (embedded via equation 22–equation 23). Define the *gain per ancilla*

$$g_e = \frac{E_S - E_{S \cup \{e\}}}{L_e}. \tag{25}$$

Greedily add the edge with largest $g_e$ while the total chain length $\sum_{f \in S} L_f$ stays within budget $K$, and stop when the best $g_e$ drops below a small threshold.

#### E.6.2 ONE-SHOT ANNEALED VALUES + KNAPSACK

Build the *base* strict 1D NN model that keeps and prunes all far edges per policy, and anneal it to get $E_\varnothing$. For each $e \in \mathbb{E}_{far}$, build the model that embeds *only* $e$ (on top of the base), anneal to get $E_{\{e\}}$, and define its annealed value:

$$v_e := E_\varnothing - E_{\{e\}} \quad (\geq 0 \text{ when helpful}). \tag{26}$$

Select $S$ by solving the 0–1 knapsack

$$\max_{y \in \{0,1\}^{|\mathbb{E}_{far}|}} \sum_{e \in \mathbb{E}_{far}} v_e\, y_e \quad \text{s.t.} \quad \sum_{e \in \mathbb{E}_{far}} L_e\, y_e \leq K, \tag{27}$$

or its QUBO encoding

$$\min_{y \in \{0,1\}^{|\mathbb{E}_{far}|}} -\sum_e v_e\, y_e + \lambda\Big(\sum_e L_e\, y_e - K\Big)^2, \tag{28}$$

with $\lambda > \max_e v_e$.

### E.7 Penalty Sweep (Choosing $M$)

Choose the smallest $M$ in a short sweep $M \in \{M_0, 2M_0, 4M_0, \dots\}$ that (a) yields $\approx 0\%$ chain breaks (equalities hold at minima) and (b) does not degrade best energy. Excessively large $M$ can drown the problem signal.

### E.8 Full Pipeline

#### Notation helpers

- Anneal$(Q, q, c; \text{reads}, \text{gauges}) \to (E_{\min}, x_{\min})$.

- BuildModel$(Q, q, c, p, S, M, \text{policy}) \to (Q', q', c', \text{ancillas})$: strict 1D NN model with kept, $S$ embedded via equation 22–equation 23, others pruned by policy.

- ChainLen$(e, p) = L_e$.

- Candidates$(\mathbb{E}_{far}, S, m)$: the $m$ far edges with largest $Q_{ij}$ not in $S$.

---

**Algorithm 2** Annealer-Guided 2D-QUBO $\to$ Strict 1D NN QUBO (Greedy Selection)

---

**Require:** $(Q, q, c)$, line order $p$, budget $K$, seed $M_0$, policy $\in \{\text{drop}, \text{fold}\}$, anneal settings
**Ensure:** Final $(Q', q', c')$, $(x_{\text{final}}, E_{\text{final}})$, selected set $S$
1: $(E^*, x^*) \leftarrow Anneal(Q, q, c; \text{reads}_{\text{ref}}, \text{gauges}_{\text{ref}})$ Reference optimum
2: Partition edges into and $\mathbb{E}_{far}$ using $p$
3: $S \leftarrow \varnothing$, $B \leftarrow 0$, $M \leftarrow M_0$
4: $(Q_0, q_0, c_0, \_) \leftarrow$ BuildModel$(Q, q, c, p, S, M, \text{policy})$
5: $(E_S, \_) \leftarrow$ Anneal$(Q_0, q_0, c_0; \text{reads}_{\text{score}}, \text{gauges}_{\text{score}})$
6: **while** $B < K$ **do**
7: $\quad \mathcal{C} \leftarrow$ Candidates$(\mathbb{E}_{far}, S, m)$
8: $\quad (\Delta^{best}, e^{best}, E^{best}) \leftarrow (0, \bot, \infty)$
9: $\quad$ **for** $e \in \mathcal{C}$ **do**
10: $\quad\quad L \leftarrow$ ChainLen$(e, p)$
11: $\quad\quad$ **if** $B + L > K$ **then**
$\quad\quad\quad$ continue
12: $\quad\quad$ **end if**
13: $\quad\quad (Q_e, q_e, c_e, \_) \leftarrow$ BuildModel$(Q, q, c, p, S \cup \{e\}, M, \text{policy})$
14: $\quad\quad (E_{S \cup \{e\}}, \_) \leftarrow$ Anneal$(Q_e, q_e, c_e; \text{reads}_{\text{score}}, \text{gauges}_{\text{score}})$
15: $\quad\quad g \leftarrow \dfrac{E_S - E_{S \cup \{e\}}}{\max(L, 1)}$
16: $\quad\quad$ **if** $g > \Delta^{best}$ **then**
17: $\quad\quad\quad (\Delta^{best}, e^{best}, E^{best}) \leftarrow (g, e, E_{S \cup \{e\}})$
18: $\quad\quad$ **end if**
19: $\quad$ **end for**
20: $\quad$ **if** $e^{best} = \bot$ or $\Delta^{best} \leq \varepsilon$ **then**
$\quad\quad$ break
21: $\quad$ **end if**
22: $\quad S \leftarrow S \cup \{e^{best}\}, \quad B \leftarrow B + ChainLen(e^{best}, p), \quad E_S \leftarrow E^{best}$
23: **end while**
24: $M \leftarrow$ PenaltySweep$(Q, q, c, p, S, \text{policy}, \{M_0, 2M_0, 4M_0, \dots\})$
25: $(Q', q', c', \_) \leftarrow$ BuildModel$(Q, q, c, p, S, M, \text{policy})$
26: $(E_{\text{final}}, x_{\text{final}}) \leftarrow$ Anneal$(Q', q', c'; \text{reads}_{\text{final}}, \text{gauges}_{\text{final}})$
27:
28: **return** $(Q', q', c')$, $(x_{\text{final}}, E_{\text{final}})$, $S$

---

---

**Algorithm 3** One-Shot Selection via Annealed Values + Knapsack

---

**Require:** $(Q, q, c)$, line order $p$, budget $K$, seed $M_0$, policy $\in \{\text{drop}, \text{fold}\}$
**Ensure:** Selected set $S$ and final model $(Q', q', c')$
 1: $(E^*, x^*) \leftarrow \text{ANNEAL}(Q, q, c)$
 2: Build base strict 1D NN model (keep , prune $\mathbb{E}_{far}$ per policy); $(E_\varnothing, \_) \leftarrow \text{ANNEAL}(\text{base})$
 3: **for** each $e = (i, j) \in \mathbb{E}_{far}$ **do**
 4:     Build "base + embed $e$ with $M_0$"; $(E_{\{e\}}, \_) \leftarrow \text{ANNEAL}$
 5:     $v_e \leftarrow E_\varnothing - E_{\{e\}};\quad L_e \leftarrow pos(i) - pos(j) - 1$
 6: **end for**
 7: Solve knapsack equation 27 (or QUBO equation 28) to get $S$
 8: $M \leftarrow \text{PENALTYSWEEP}(Q, q, c, p, S, \text{policy}, \{M_0, 2M_0, 4M_0, \dots\})$
 9: Build final strict 1D NN model (embed all $e \in S$ with $M$; keep ; prune others)
10: **return** $(Q', q', c')$, and then $(x_{\text{final}}, E_{\text{final}}) \leftarrow \text{ANNEAL}(Q', q', c')$

---

# F    TIME COMPLEXITY OF ISING SOLVERS: FORMAL MEASURES AND $O(\cdot)$ SCALINGS

**Problem setting.**    Let $G = (V, E)$ be a graph with $|V| = n$ spins $s_i \in \{\pm 1\}$, couplings $J_{ij}$, and (optional) fields $h_i$. We minimize the Ising energy

$$E_{\text{Ising}}(s) = -\tfrac{1}{2} \sum_{i,j} J_{ij} s_i s_j - \sum_i h_i s_i. \tag{29}$$

Computing a ground state is NP-hard in general (e.g., for 3D lattices, and for planar graphs with fields) Barahona (1982).

## F.1    WHAT WE MEAN BY "TIME COMPLEXITY" FOR HEURISTIC ISING SOLVERS

For solvers whose single run of duration $\tau$ succeeds with probability $p(\tau)$ on a given instance, a device/algorithm–agnostic metric is the *time–to–solution (TTS)* for target confidence $1 - \delta$:

$$_{1-\delta} = \min_{\tau > 0} \tau \frac{\log \delta}{\log(1 - p(\tau))}. \tag{30}$$

This is the standard "repeat–until–success" model used in annealing benchmarks; the optimization over $\tau$ emphasizes that *per-size* schedules must be tuned Rønnow et al. (2014). For Monte Carlo algorithms we count time in *Monte Carlo sweeps* (MCS; cf. one attempted update per spin), then convert to wall time by the measured seconds per sweep.

Two analytic lenses connect equation 30 to exact mathematical quantities. First, for reversible Markov chains (SA and SQA at fixed control parameter), the *mixing time* $t_{\text{mix}}(\varepsilon)$ obeys

$$t_{\text{mix}}(\varepsilon) \leq \frac{1}{\gamma} \log \frac{1}{\varepsilon \, \pi_{\min}}, \tag{31}$$

where $\gamma$ is the spectral gap and $\pi_{\min}$ is the minimum stationary probability (Levin et al., 2017, Thm. 20.6). Second, for closed-system QA (adiabatic evolution) the runtime $T$ needed for error $\leq \varepsilon$ satisfies the gap-controlled bound

$$T = O\left( \frac{\max_{s \in [0,1]} \left\| \frac{dH}{ds} \right\|}{\varepsilon \, \Delta_{\min}^2} \right), \tag{32}$$

with $\Delta_{\min}$ the minimum instantaneous spectral gap along the anneal path $H(s)$ Jansen et al. (2007); Albash & Lidar (2018).

## F.2    ALGORITHM-BY-ALGORITHM FORMULATIONS

**(1) Simulated Annealing (SA).**    SA runs a temperature-dependent (often single-spin Metropolis) Markov chain with a cooling schedule $T_k \downarrow 0$. A precise convergence criterion is given by Hajek: if

$$T(k) = \frac{c}{\log(1 + k)} \quad \text{with} \quad c \geq d^*, \tag{33}$$

then the algorithm converges in probability to the set of global minima; here $d^*$ is the *depth* of the deepest non-global local minimum (a barrier-height functional) Hajek (1988).

At fixed temperature $T$, the chain's mixing time obeys equation 31; running an anneal that stays near equilibrium yields the upper bound

$$\underset{1-\delta}{\text{SA}} = O\left(\sum_{k=1}^{K^*} \frac{1}{\gamma(T_k)} \log\left(\frac{1}{\varepsilon_k \, \pi_{\min}(T_k)}\right)\right),$$

(34)

for a quasi-static schedule $\{T_k\}_{k \leq K^*}$ and per-stage accuracies $\varepsilon_k$ determined by the schedule discretization.[1]

*Empirical example.* On random Chimera graphs with $\pm 1$ couplings and optimized anneal times, SA exhibits $= \exp(\Theta(\sqrt{N}))$ scaling (Rønnow et al., 2014, Fig. 3).

**(2) Simulated Quantum Annealing (SQA).** SQA uses path-integral QMC to sample the Gibbs state of a stoquastic transverse-field Ising Hamiltonian; the Markov chain evolves on an expanded state space of $n \times L$ spins (with $L$ Trotter slices). At a fixed $(\beta, \Gamma)$, the *PIMC* chain's mixing time $t_{\min}^{\text{PIMC}}$ controls runtime:

$$\underset{1-\delta}{\text{SQA}} = O\left(t_{\text{mix}}^{\text{PIMC}}(n, \beta, \Gamma) \cdot \log \tfrac{1}{\delta}\right), \qquad \text{per MCS cost} = O(\text{nnz}(G)\, L),$$

(35)

where $\text{nnz}(G)$ counts spin–spin couplers actually touched per sweep. There are families where $t_{\text{mix}}^{\text{PIMC}}$ is *provably polynomial*, e.g., 1D stoquastic Hamiltonians at $\beta = O(\log n)$ Crosson & Harrow (2021), and the "spike" cost function where SA is exponentially slow but SQA finds the optimum in $\text{poly}(n)$ time Crosson & Harrow (2016). *Empirical example.* On the same random Chimera benchmarks as above, optimized SQA also exhibits $= \exp(\Theta(\sqrt{N}))$ scaling Rønnow et al. (2014).

**(3) Quantum Annealing (QA; adiabatic algorithm / analog devices).** For a Hamiltonian path $H(s) = (1-s)H_0 + sH_1$, the adiabatic theorem yields the gap-based complexity in equation 32. Thus,

$$\boxed{\underset{1-\delta}{\text{QA}} = O\left(\frac{\|\dot{H}\|_\infty}{\varepsilon \, \Delta_{\min}^2}\right) \quad \text{(ideal closed system)}}$$

(36)

(up to smoothness constants), so exponentially small $\Delta_{\min}$ implies exponential time. There are random NP-complete families (e.g., Exact Cover) where $\Delta_{\min}$ is exponentially small with high probability, hence adiabatic QA requires exponential time Altshuler et al. (2010). For hardware experiments, one computes TTS exactly as in equation 30 using the per-run anneal time and observed success $p$ Rønnow et al. (2014).

**(4) Simulated Bifurcation (SB).** SB integrates a classical time-dependent Hamiltonian flow on variables $(x_i, y_i)$ with forces

$$f_i = \sum_j J_{ij}\, u_j, \qquad u_j = \begin{cases} x_j & \text{(ballistic SB)}, \\ \text{sgn}(x_j) & \text{(discrete SB)}. \end{cases}$$

(37)

Per time step, the dominant arithmetic is applying $J$ to $u$, i.e.,

$$\text{cost/step} = O(\text{nnz}(J)) \quad \text{(sparse)} \quad \text{or} \quad O(n^2) \quad \text{(dense)}.$$

(38)

SB papers report a *step–to–solution* (StS) metric that is independent of machine speed: if $N_s$ steps are taken per trial and $P$ is the per-trial success probability, then

$$\boxed{\text{StS}_{1-\delta} = N_s \frac{\log \delta}{\log(1-P)}}, \qquad \underset{1-\delta}{\text{SB}} = \text{StS}_{1-\delta} \times \text{(time/step)}.$$

(39)

Heated/discrete variants reduce StS on Sherrington–Kirkpatrick (dense) benchmarks Kanao & Goto (2022); see Goto et al. (2019; 2021) for the original dynamics and high-performance variants.

---

[1] In the worst case $d^*$ can grow with $n$, so schedules meeting equation 33 can imply super-polynomial runtime on typical NP-hard families; cf. Barahona (1982).

## F.3 SIDE-BY-SIDE SUMMARY (DOMINANT TERMS)

| Method | Asymptotic lens for $1-\delta$ |
| --- | --- |
| SA | $O\left(\sum_k \gamma(T_k)^{-1} \log \frac{1}{\varepsilon_k \pi_{\min}(T_k)}\right)$ for a quasi-static schedule; converges if $T(k) = c/\log(1+k)$ with $c \geq d^*$ (barrier depth) Hajek (1988); Levin et al. (2017). |
| SQA | $O\left(t_{\mathrm{mix}}^{\mathrm{PIMC}}(n, \beta, \Gamma) \cdot \log \frac{1}{\delta}\right)$ with per-sweep $O(\mathrm{nnz}(G)L)$; polynomial on certain families (e.g. 1D stoquastic, spike). Crosson & Harrow (2021; 2016) |
| QA | $O\left(\|\dot{H}\|_\infty / (\varepsilon \Delta_{\min}^2)\right)$ (adiabatic bound); small gaps $\Rightarrow$ exponential time. Jansen et al. (2007); Albash & Lidar (2018) |
| SB | $\mathrm{StS}_{1-\delta} \times$ (time/step), with StS in equation 39 and cost/step in equation 38. Kanao & Goto (2022); Goto et al. (2019) |

**Practical note.** When reporting complexity empirically, optimize schedule/parameters *per* size $n$ and quote $1-\delta$ from equation 30; failure to re-optimize can mask speedups/slowdowns Rønnow et al. (2014).

