# OpenReview forum: "THE BLACK–WHITE-BOX OPTIMIZATION NETWORK"
_ICLR.cc/2026/Conference — ICLR 2026 Conference Withdrawn Submission_

### Official Review · Reviewer_ESWM · 2025-10-31

**Soundness:** 3
**Presentation:** 3
**Contribution:** 3
**Rating:** 6
**Confidence:** 3

**Summary:**

The paper proposes a new hybrid optimization framework called the Black–White-Box Optimization Network (BWBON), aiming to bridge black-box and white-box optimization paradigms. The authors introduce its first concrete implementation, the Tensor-Train Creator (TTC). The system operates iteratively: white-box Ising solves generate “seeds” that warm-start the TT-based sampler, while the surrogate helps realign the optimization lattice to reduce tensor ranks and accelerate convergence. The authors formalize this process as a typed solver graph that can mix different optimization modules through explicit couplings.

**Strengths:**

1. This paper moves beyond simple grey-box or two-module hybrids toward a general multi-solver architecture.
2. This paper provides insights into why dense QUBO problems are difficult for TT methods and how TTC mitigates this through lattice alignment.
3. This paper includes formal complexity analysis and convergence-related theorems with explicit scaling laws.

**Weaknesses:**

1. Experiments mainly focus on synthetic benchmarks and Max-Cut and no large-scale or real-world application is included.
2. While annealer and surrogate costs are discussed, actual wall-clock trade-offs are not shown experimentally.

**Questions:**

1. Could the method generalize to continuous or mixed-integer optimization problems beyond discrete QUBO/HUBO formulations?
2. What is the empirical cost of the annealing-based shaping step compared to the overall optimization time?

---

### Official Review · Reviewer_NpkF · 2025-10-31

**Soundness:** 1
**Presentation:** 1
**Contribution:** 1
**Rating:** 0
**Confidence:** 3

**Summary:**

This paper proposes Tensor-Train Creator (TTC), a hybrid framework for solving discrete optimization problems such as Max-Cut. TTC integrates three components: a Higher-Order Factorization Machine (HOFM) for modeling polynomial structures, a quantum-like solver for initialization, and PROTES, a tensor-train–based probabilistic search. The framework aims to unify white-box (structured) and black-box (oracle-based) optimization through tensorized “typed couplings.” Some theoretical motivation and limited experimental results are presented to suggest potential efficiency gains.

**Strengths:**

The proposed algorithm (Algorithm 1) is well laid out and conceptually clear in showing how TTC integrates surrogate modeling, annealing-based seeding, and tensor-train sampling.

**Weaknesses:**

The paper is extremely difficult to follow. It lacks a clear and consistent problem formulation and precise definitions of basic concepts (e.g., what exactly is meant by “typed coupling” or “black–white box coupling”). Important terms like QUBO, HUBO, and TT decomposition appear without prior introduction or references for non-expert readers. For example, equation (3) uses a nonstandard inner product notation ⟨v_{i_1}^{(k)}, …, v_{i_k}^{(k)}⟩ that is never explicitly defined. These issues make the manuscript inaccessible even for an informed reader familiar with tensor optimization.

The manuscript mixes multiple ideas—Ising solvers, factorization machines, TT representations, QUBO/HUBO transformations—without providing sufficient intuition or connection among them. It is unclear what specific problem TTC is solving and how the submodules interact at each iteration in a mathematically rigorous way. For instance, PROTES and HOFM are each described in detail, but the way they exchange information is mostly described narratively rather than formally.

The numerical section (Table 2) is very minimal, lacking experimental setup details such as dataset sizes, number of evaluations, parameter settings, and runtime comparison methodology. No standard deviation or error metrics are given. This makes it impossible to assess reproducibility or robustness. The claim of “better values under the same evaluation budgets” is not substantiated with statistical evidence or ablation studies.

While the architecture combines several existing techniques, it is unclear whether the observed improvements come from the combination itself or from specific hyperparameter tuning. There is no clear ablation showing the effect of each component (e.g., HOFM alone vs. PROTES alone vs. TTC). Moreover, the relationship between the theoretical discussion in Section 4 (QUBO vs. HUBO) and the practical implementation in Algorithm 1 is not clearly established.

The paper includes many formal statements and theorems that appear mathematically correct but are not directly validated or tied to the main contribution. Many results are re-statements of known properties of tensor-train decompositions and treewidth, yet presented as novel insights.

**Questions:**

The paper is evidently not yet ready for submission. It requires major revisions to improve structure, clarity, and experimental rigor. Therefore, I have no specific technical questions to ask the authors at this stage.

---

### Official Review · Reviewer_4brZ · 2025-10-31

**Soundness:** 2
**Presentation:** 1
**Contribution:** 2
**Rating:** 2
**Confidence:** 3

**Summary:**

The paper introduces a Black–White-Box Optimization Network and instantiates it as the Tensor-Train Creator (TTC): a hybrid pipeline that interleaves an Ising/annealing “white-box” path with a tensor-train (TT)–based “black-box” search, using a surrogate (HOFM) to reveal low-rank structure and guide a TT sampler (PROTES). TTC warm-starts TT from annealer-derived solutions and realigns problem structure to keep TT ranks small, reducing oracle evaluations and time-to-target compared to standalone black-box or annealing methods. The paper also analyzes when TT methods struggle on dense QUBO and how structured QUBO→HUBO transformations can help, and reports empirical improvements on synthetic benchmarks and Max-Cut.

**Strengths:**

1. Clear modularity and novelty of composition. The “typed couplings” idea crisply formalizes how white- and black-box solvers exchange information

2. Useful theory explaining TT behavior. The lattice-alignment result (treewidth/separator view) convincingly explains why dense QUBO induces large TT ranks and why certain HUBO re-embeddings help; this is a valuable conceptual contribution for TT-based optimizers.

**Weaknesses:**

1. Experimental evidence appears light for an ICLR-level systems claim. While the abstract reports wins under equal budgets, the paper would benefit from fuller methodology: dataset/task details, variance across seeds, wall-clock vs. evaluation-count breakdowns, and stronger ablations (e.g., HOFM-only shaping, annealer-only warm-start, PROTES-only with tuned ranks). The current text does not yet show broad, statistically robust coverage commensurate with the breadth of claims.

2. Reliance on annealing backends. TTC’s advantage hinges on (i) quality/latency of Ising solves and (ii) faithful QUBO extraction from HOFM; the paper acknowledges annealer overheads but more empirical quantification (TTS, reads, gauges) is needed to assess portability across devices and settings.

3. Surrogate fidelity and stability. Restricting to the quadratic part for QUBO construction can misrepresent higher-order effects that HOFM models; guidance on when this truncation helps/hurts (and sensitivity to \lambda/penalties) is limited.

4. Scope of the HUBO construction. The constructive “near-chain” HUBO argument is compelling, but the paper stops short of giving automatic pipelines with approximation guarantees or empirical stress-tests on real-world dense instances.

5. clarity and organization. The writing makes it difficult to clearly separate new contributions from prior methods. The technical narrative often mixes background, intuition, and algorithmic details without a clean progression. A clearer structure (e.g., “background → key insights → method → theory → experiments”) and explicit novelty call-outs would strengthen readability and impact.

**Questions:**

1. TT rank guarantees: The theory explains when TT will fail on dense QUBO, but do you have formal bounds or probabilistic guarantees for success in the structured settings chosen?

2. Why is the quadratic truncation of HOFM used for QUBO extraction? Is there empirical evidence that higher-order terms harm performance, or is this primarily for tractability?

3. Ablations: Can you provide ablations to identify the contribution of each part? e.g,TT alone; TT+warm-start; surrogate-only shaping; TT + HUBO transformation without annealing

4. Can you provide a detailed breakdown of wall-clock runtime, number of oracle calls, and annealer time? How sensitive are results to the relative cost of annealing?

---

### Official Review · Reviewer_Fp4i · 2025-11-04

**Soundness:** 3
**Presentation:** 1
**Contribution:** 2
**Rating:** 2
**Confidence:** 2

**Summary:**

This paper formalizes the black-white-box optimization network and proposes the Tensor-Train Creator (TTC) for general optimization problems. The TTC framework interleaves black-box and white-box optimization to reap the benefits of both techniques. Specifically, TTC is composed of a higher-order factorization machine, an ising-machine that solves a component of the output of the HOFM (the white-box solver), and a Tensor-Train based black-box solver. Experiments show that under the same budget, TTC outperforms other baselines on several black-box optimization problems.

**Strengths:**

1. Novelty: the TTC framework is novel as far as I know, combining several components such as the Ising-machine and HOFM. Some of these components, such as the Ising-machine, can have a quantum backend which I find interesting.
2. The paper gives a detailed analysis of the complexity of the procedure, and outlines when there is an advantage for using TTC compared to other methods.
3. Several optimization experiments demonstrate that TTC outperforms other baselines under the same budget.

**Weaknesses:**

1. My main concern is writing. The paper describes an optimization procedure, which is of broad interest to the ML community, but uses many terminologies that are not formally introduced in the paper with sufficient background. For example, what's QUBO and HUBO? On the first page, in the definition of HOFM, what are the w's and v's?

2. Why does the Ising solver only solve the quadratic problem, instead of a higher-order problem? Does performance improve if we retain higher order factorizations?

3. To clearly show the benefits of TTC, could the authors provide a breakdown of the cost of running TTC for the experimental benchmarks? For example, how much compute was spent in fitting the HOFM and solving the QUBO? It would also be interesting to quantify the benefit of warm-starts, and one potential experiment is to compare the quality of the points evaluated with and without warm-starts.

4. Section 5 is missing exposition: what are the dimensions of these optimization problems, and what's the fixed budget in evaluating them? Is the budget fixed in terms of compute or number of evaluations? What are BS1 and BS2, and are they SOTA solvers for these problems?

**Questions:**

See weaknesses.

---

### Note · Authors · 2025-11-12

I have read and agree with the venue's withdrawal policy on behalf of myself and my co-authors.